# Indications for Sleeve Gastrectomy—Is It Worth Waiting for Comorbidities to Develop?

**DOI:** 10.3390/medicina59122092

**Published:** 2023-11-29

**Authors:** Zsuzsanna Németh, Miklós Siptár, Natália Tóth, Krisztina Tóth, Csaba Csontos, Zoltán Kovács-Ábrahám, Alexandra Csongor, Ferenc Molnár, Zsombor Márton, Sándor Márton

**Affiliations:** 1Medical Skills Education and Innovation Centre, Medical School, University of Pécs, 7624 Pécs, Hungary; ferenc.molnar@aok.pte.hu; 2Southern Transdanubian Region, National Ambulance Service, 1055 Budapest, Hungary; 3Department of Anaesthesiology and Intensive Therapy, Medical School, University of Pécs, 7624 Pécs, Hungary; siptar.miklos@pte.hu (M.S.); toth.natalia2@pte.hu (N.T.); toth.krisztina2@pte.hu (K.T.); csaba.csontos@gmail.com (C.C.); kovacs-abraham.zoltan@pte.hu (Z.K.-Á.); marton.sandor@pte.hu (S.M.); 4Department of Languages for Biomedical Purposes and Communication, Medical School, University of Pécs, 7624 Pécs, Hungary; alexandra.csongor@aok.pte.hu; 5Department of Anatomy, Medical School, University of Pécs, 7624 Pécs, Hungary; zsombor.marton@aok.pte.hu

**Keywords:** obesity, obesity surgery, bariatric surgery, gastrectomy, comorbidity, hypertension remission, obstructive sleep apnea, diabetes mellitus

## Abstract

(1) *Background and Objectives:* Morbid obesity significantly increases the prevalence of comorbidities, such as heart disease, restrictive lung disease, stroke, diabetes mellitus and more. (2) *Methods*: Patients undergoing gastric sleeve surgery were divided into three groups with BMI between 30–34.9 kg/m^2^ (Group I), 35–39.9 kg/m^2^ (Group II), and over 40 kg/m^2^ (Group III). Preoperative examinations included cardiac ultrasound, respiratory function and laboratory tests, and preoperative comorbidities were also recorded. Following a one-year follow-up, we compared the rate of weight loss in the three groups at six months and one year following surgery, specifically, the effect of surgery on preoperative comorbidities at one year. (3) *Results*: The weight loss surgeries performed were successful in all three groups. Preoperative laboratory examinations, an echocardiogram and respiratory function results showed no clinically significant difference, except moderate elevations in blood lipid levels. Hypertension was the most common comorbidity. (4) *Conclusions*: In our patient population, hypertension and diabetes were the only comorbidities with a high prevalence. It can be explained by the relatively younger age among the patients (mean age 44.5 years) and the fact that they had not yet developed the pathological consequences of severe obesity. Consequently, while performing the surgery at a relatively younger age, it seems far more likely that the patient will return to a more active and productive life and enjoy a better quality of life. Additionally, the perioperative risk is lower, and the burden upon health systems and health expenditure is reduced by preventing comorbidities, in particular, multimorbidity. On this basis, it may be advisable to direct patients who do not exhaust the classical indications for bariatric surgery toward the surgical solution at a younger age. Our results suggest it is not worth waiting for comorbidities, especially multimorbidity, to appear.

## 1. Introduction

The World Health Organization (WHO) has declared obesity as the largest global chronic health problem among adults, which is increasingly turning into a more serious problem than when compared with malnutrition. Obesity is a gateway to ill health, and it has become one of the leading causes of disability and death, affecting not only adults but also children and adolescents worldwide [1].

Obesity is a growing challenge in our modern society. Its prevalence in the United States of America was 30.5% in 1999–2000, rising to 42.4% in 2017–2018, while the prevalence of severe obesity increased from 4.7% to 9.2% over the same period [2]. Based on current trends, the proportion of overweight individuals in the US is expected to rise to 48.9% by 2030 [3], i.e., in which one in two Americans are estimated to be obese.

Obesity results from a chronic energy imbalance characterized by a steady and increased energy intake. Biological (including genetic and epigenetic), behavioral, social, and environmental factors (including chronic stress) regulate energy balance and fat stores. The rapid increase in the prevalence of obesity over the past 30 years is primarily the result of cultural and environmental influences. High energy-dense diets, low physical activity, sedentary lifestyles and eating disorders are important risk factors for obesity [4].

Morbid obesity occurs when the body’s relative or absolute fat content increases. In discussing obesity, it is essential to introduce the concept of Body Mass Index (BMI), the ratio of body weight in kilograms divided by the square of height in meters. BMI provides the most useful population-level measure of overweight and obesity as it is the same for both sexes and for all ages of adults. However, it should be considered a rough guide since it may not correspond to the same degree of fatness among different individuals [5]. If the BMI is below 18.5 kg/m^2^, the patient is abnormally thin; if it is between 18.5 and 24.9 kg/m^2^, the patient is of average weight; between 25 and 29.9 kg/m^2^, the patient is overweight; and if the BMI is above 30 kg/m^2^, the patient is obese, which includes Class I (30–34.9 kg/m^2^), Class II (35–39.9 kg/m^2^) and Class III (over 40 kg/m^2^) obesity [6]. There are two basic patterns of adipose tissue accumulation. The gynoid type of obesity (also known as peripheral, subcutaneous, gluteofemoral, or pear-type obesity) is known to be more common among females, while the android (or centric, visceral, abdominal, appendicular, apple) type of obesity is known to be more common in males. However, both types can occur in both sexes [7]. At the same time, according to the 2014 Hungarian Diet and Nutritional Status Survey, abdominal obesity is less common in males than in females (38% vs. 55%) [8]. Android obesity is characterized by increased deposition of fat in the thorax, abdomen and visceral organs, of which gynoid obesity is characterized by deposition of fat in the hips, thighs and limbs, and in subcutaneous tissue. This distinction is important. Android obesity is likely to have a more direct effect on pulmonary mechanics; it also bears a greater impact on metabolic inflammation. Visceral fat is more metabolically and hormonally active than subcutaneous fat. Indeed, increased visceral fat mass is linked to metabolic syndrome, and metabolic syndrome has been linked to asthma and impaired lung function in both adolescents and adults [9].

Primary obesity develops when patients consistently take in more calories than they burn. Secondary obesity is a complication of some diseases.

Contrary to solely being a medical condition or risk factor for other diseases, obesity is a complex disease of multifaceted etiology, with its own disabling capacities, pathophysiologies and comorbidities. It meets the medical definition of disease in which it is a physiological dysfunction of the human organism with environmental, genetic and endocrinological etiologies [10]. 

Obesity is a systemic issue and, as a result, affects multiple different organ systems. The associated conditions range from poor mood and skin infection to potentially life-threatening [11], including hypertension, coronary artery disease, heart failure, atrial fibrillation, stroke, venous thromboembolism, obstructive sleep apnea and obesity hypoventilation syndrome, Type 2 diabetes mellitus, dyslipidemia, metabolic syndrome, vitamin D deficiency, idiopathic intracranial hypertension, osteoarthritis [11], gastroesophageal reflux and associated hiatal hernias [12], non-alcoholic fatty liver disease, gallstone, urinary and reproductive tract disorders, psychiatric disorders, integumentary system disorders, infections and neoplasms. Moreover, obese individuals more commonly experience falls and fractures [11].

Increased body weight increases the body’s oxygen demand (VO_2_). It, therefore, increases the effort in breathing, decreases thoracic compliance, leading to the development of restrictive lung disease, and decreases expiratory reserve volume and functional residual capacity; thus, the ventilation-perfusion ratio becomes unfavorable [13]. The circulating blood and plasma volumes are high, hence the minute volume, which increases the left ventricular size and end-diastolic pressure, ultimately leading to left heart failure. Due to the higher pulmonary blood volume, pulmonary hypertension is expected, which can lead to right ventricular failure. These effects can combine to produce biventricular failure. High serum levels of blood lipids are often observed, which may exacerbate any ischemic heart disease. Glucose intolerance is frequent, highlighted by hypertrophy of pancreatic cells, consequently leading to diabetes.

Obesity is the leading risk factor in the development of Type 2 diabetes. Females with a BMI of over 30 kg/m^2^ bear a 28-fold increased risk of developing diabetes compared with females of average body mass and a 93-fold increased risk for females with a BMI of over 35 kg/m^2^ [14]. Obese females have a higher rate of infertility, miscarriage, polycystic ovary syndrome, higher rates of cesarean section, placental perfusion defects and fetal distress [15] and also embryonic developmental defects and abnormality in their offspring [16]. In the case of males, the complications of obesity include erectile dysfunction, poor semen quality and subclinical prostatitis; therefore, obesity is known to disrupt male fertility and the reproduction potential, particularly through alteration in the hypothalamic-pituitary-gonadal axis, disruption of testicular steroidogenesis and metabolic dysregulation, including insulin, cytokines and adipokines [17]. 

There is convincing evidence suggesting excess body weight is associated with an increased risk for cancer of at least 13 anatomic sites, including endometrial, esophageal, renal and pancreatic adenocarcinomas; hepatocellular carcinoma; gastric cardia cancer; meningioma; multiple myeloma; colorectal, postmenopausal breast, ovarian, gallbladder and thyroid cancers [18].

Morbidity and mortality rates are higher in morbidly obese patients with acute morbidities such as polytrauma, sepsis, or severe burns. Additionally, obesity was found to be a significant risk factor for intensive care unit admission, including invasive mechanical ventilation in COVID-19 infection. In comparing body mass index classes with one another, a higher BMI was consistently associated with higher risks [19]. 

There are two treatment options for obesity: conservative and surgical. The two pillars of the conservative approach are increasing physical activity and reducing calorie intake. Surgical treatment may be justified if conservative methods fail unless secondary obesity is confirmed.

The classic criteria in order for a patient to be a candidate for any bariatric surgery is a BMI score greater than or equal to 40 kg/m^2^ or a BMI of 35–40 kg/m^2^ with at least one obesity-related comorbid condition, such as hypertension, diabetes mellitus, or severely limiting musculoskeletal issues (near unsuccessful nonoperative weight loss attempts, mental health clearance, no medical contraindication to surgery). Recent updates now include patients with a BMI of 30–35 kg/m^2^ with uncontrollable Type 2 diabetes or metabolic syndrome as an indication for a laparoscopic sleeve gastrectomy [20,21].

Currently, several types of bariatric surgery are recognized. The action mechanism of these surgeries can be restrictive, in which the aim is to limit food intake, or malabsorptive, in which the aim is to reduce absorption. Generally speaking, restrictive procedures are more effective, yet the durability of the effect may be less than when compared to malabsorptive procedures. 

According to 2020 data, LSG remains the most common bariatric surgery in the United States [22]. In addition to its restrictive effect, the procedure is also referred to as an endocrine surgical procedure since the gastric fundus is removed, in which a majority of ghrelin-producing cells are located [23]. Among the many metabolic effects of ghrelin, the most notable is the increase in appetite and the stimulation of lipogenesis independent of food intake, which leads to an increase in body weight and obesity [24].

## 2. Materials and Methods

Based on the principles of the Declaration of Helsinki, approved by the Ethics Committee of the University of Pécs, Faculty of Medicine (permission number: 8653—PTE 2021), with the appropriate registration of the study (Clinical Trials.gov, Identifier: NCT05929170), and following written informed consent among the patients, we studied the data of 164 patients (98 females and 66 males) who underwent laparoscopic sleeve gastrectomy between 2021 and 2022. All patients had a BMI higher than 30 kg/m^2^, were aged between 18 and 70 years old and had not undergone previous bariatric surgery. Our study was a single-center follow-up study in which our patient population consisted only of individuals from Hungary. Additionally, all operations were performed by the same surgeon.

Patients underwent preoperative investigations, including consultation with a surgeon, an anesthesiologist, an internist, internal medicine examination and medication adjustment, gastroscopy, psychological examination, cardiac ultrasound, respiratory function tests, laboratory tests and preoperative comorbidities were recorded, including anthropometric data.

All patients’ sex, age, height and weight prior to surgery, BMI, ideal body weight and body fat percentages were collected. We also recorded the patients’ preoperative laboratory parameters, such as total cholesterol, high-density lipoprotein (HDL), triglyceride, low-density lipoprotein (LDL), thyroid stimulating hormone (TSH), fasting blood glucose, hemoglobin A1c (Hgb A1c), morning cortisol levels, etc. Patients also underwent cardiac ultrasound (diastolic left ventricular diameter, ejection fraction, right ventricular diameter, E/A ratio) and individualized preoperative pulmonary function values. These include FVC (Forced Vital Capacity), Forced Expiratory Volume (FEV1), Peak Expiratory Flow (PEF) and the Tiffeneau Index. Additionally, pre- and post-operative tobacco use was reported.

The possible comorbidities that frequently accompany obesity were also noted, such as hypertension, diabetes, impaired fast glucose, impaired glucose tolerance, insulin resistance, asthma, gastroesophageal reflux disease, hypothyroidism, anxiety, polycystic ovarian syndrome, obstructive sleep apnea syndrome (OSAS) and gout.

We evaluated the procedure’s effectiveness six months and one year following surgery regarding weight loss, of which we noted the percentage reduction in BMI and improvement or complete disappearance of comorbidities preoperatively present at the same time points.

In consideration of OSAS, we reviewed those patients who were screened at the appropriate center and diagnosed with OSAS—and, if necessary, undergoing therapy.

The six-month and one-year data were collected partly live, in person, and partly through telephone consultations.

Improvement of comorbidities or complete recovery from a given comorbidity was defined as a reduction in the medication administered for a given disease (discontinuation or reduction in the dose of a drug or drugs used to treat a given comorbidity) or complete discontinuation of medication administered for a given comorbidity, as prescribed by an appropriate specialist: cardiologist, internist, general practitioner).

We divided 164 patients who underwent laparoscopic sleeve gastrectomy into three groups. Members of Group I had a BMI between 30–34.9, Group II between 35–39.9 and Group III above 40 kg/m^2^. Next, we examined whether there is evidence of differences between groups in mean age, comorbidities, preoperative laboratory parameters, cardiac ultrasound findings, or whether surgery proved effective over the span of a half-year and one year regarding weight loss and improvement in comorbidities in each group.

Patients were anesthetized with pantoprazole and midazolam premedication as standard with propofol and fentanyl induction; atracurium was used for muscle relaxation, and intubation was performed with a video laryngoscope. Desflurane was used to maintain anesthesia. NSAIDs were administered for postoperative analgesia.

Data were analyzed using IBM Statistic software version 20.0, and the Spearman’s correlation analysis and Kruskal-Wallis test were used for statistical analysis. Data are presented as median and standard deviation. A *p*-value < 0.05 was considered statistically significant.

## 3. Results

Patient demographics are listed in Table 1. Significant reductions in BMI were seen in all three groups at six months and one year, demonstrating the effectiveness of surgery. The one-year BMI reduction was 32.3% in Group I, 29.8% in Group II and 34.4% in Group III. The most significant weight loss in all three groups occurred at six months *p* < 0.01, with a lower rate of weight loss between six months and one year. The rate of BMI reduction in the corresponding periods showed no statistically verifiable difference between groups. The Kruskal-Wallis test was used for statistical analysis (*p* > 0.05) (Figure 1). 

Prior to the operation, patients underwent a respiratory function test, the results of which are summarized in Figure 2. Among patients, the parameters we assessed (FVC, FEV1, PEF and Tiffeneau Index) were lowest in Group III; however, based on the results of the Kruskal-Wallis test, the difference does not reach statistical significance between the three groups, and the measured results were within the physiological range (*p* > 0.05).

The results of the cardiac ultrasound scan are summarized in Table 2. The preoperative findings were also inside the physiological range; however, the right ventricular diameter difference reached statistical significance between the three groups according to the Kruskal-Wallis test (*p* = 0.009) (The right ventricular diameter was higher in Group III compared to Groups I and II).

Our preoperative laboratory values, summarized in Table 3, were also within the physiological range, with the exception of moderate elevations in blood lipid levels. 

Among the comorbidities, hypertension was the most common ailment seen in patients we studied (Table 4); however, no significant difference was found in the incidence of hypertension among patients in the three study groups, likely due to one of the indications for surgery in Group I was hypertension as a comorbidity. The prevalence of hypertension in Groups I, II, III was 51, 43 and 45%, respectively. Additionally, patients were considered hypertensive if they were receiving antihypertensive treatment with blood pressure below 140/90 mm Hg or if their blood pressure was consistently above 140/90 mm Hg, although it should be noted, in the absence of an internal medicine investigation, our patients have not undergone surgery since, if secondary obesity is diagnosed. Treating the underlying condition causing obesity is of primary concern.

Consequently, all hypertensive patients were diagnosed with hypertension at the latest during the perioperative check-up. Patients we researched underwent surgery with appropriate antihypertensive medication and adequate blood pressure readings; thus, we did not have any persistent hypertension above 140/90 mm Hg in our follow-up. 

In all three groups studied, we observed improvement in hypertension in a significant proportion of hypertensive patients in both the six-month and one-year follow-up examinations due to weight loss following surgery and lifestyle changes (36%, 34% and 40% of hypertensive patients in Group I, II, and III, respectively), i.e., the appropriate specialist reduced the patients’ blood pressure medications. The complete abandonment of antihypertensive medications in our study period was apparent, but only for a few patients.

The prevalence of diabetes was lower than expected, with the highest prevalence in Group III at 15% (11% for Group I and Group II patients). The difference does not reach statistical significance between the groups regarding the prevalence of diabetes.

OSAS occurred in only a few patients in the population we studied, but as mentioned in the methods section above, only diagnosed patients were considered positive for OSAS. With appropriate specialist testing, presumably, many patients will have some severity of OSAS, given that obesity is a significant risk factor for OSAS. In the study by Sareli et al., 342 patients evaluated for bariatric surgery underwent overnight polysomnography and completed questionnaires regarding sleepiness, menopausal status and respiratory symptoms related to OSAS, in which they found the prevalence of OSAS in all patients considered for bariatric surgery was greater than 77%, irrespective of OSAS symptoms, gender, menopausal status, age or BMI [25]. For all the above reasons, our data are undoubtedly not presently relevant for OSAS, and further research with a methodology focused on OSAS is required.

Based on our study, we conclude that the increase in body weight is not related to the age of our patients, i.e., patients may already suffer from very severe obesity at a younger age, according to the results of Spearman’s correlation analysis (*p* = 0.082). Interestingly, of the three groups studied, Group III patients had the lowest average age.

## 4. Discussion

During our preoperative check-up, we found physiological values in all three groups, including respiratory function, cardiac ultrasound and laboratory tests (except for moderate elevation in blood lipid levels). 

Apart from hypertension and diabetes, the comorbidities typically associated with obesity did not develop in significant proportions among our patients. Surprisingly, even diabetes was confirmed in only 15% of our Group III patients, while it proved lower in Group I and II patients.

In consideration of hypertension, our data, compared with international data, showed a lower prevalence of hypertension among patients undergoing LSG (69% according to The SLEEVEPASS Randomized Clinical Trial [26], in contrast to nearly 50% in our data.)

In regards to improvement in hypertension, our results were below those published in the literature. In the meta-analysis noted by Buchwald et al., the age of patients was similar to the average age of our patients (39 years in the study), and among 22,094 bariatric surgery patients, hypertension was resolved in 61.7% of the cases and resolved or improved in 78.5% [27] of the population, while in our study one year following surgery, in nearly 40% of hypertensive patients (36% in Class I, 34% in Class II and 40% in Class III, respectively) hypertension improved, i.e., their antihypertensive therapy was reduced yet its complete discontinuation was possible only in the case of a few patients.

In consideration of Type 2 diabetes mellitus, based on the study of Vetter et al., nearly 30% of patients who undergo bariatric surgery are afflicted with Type 2 diabetes mellitus [28].

In terms of weight loss, based on the analysis of the data collected from the population among our patients who underwent LSG surgery, we conclude that surgery is effective with respect to weight reduction over the short term (one-year follow-up). The reduction in BMI at one year was 32.3% in Group I, 29.8% in Group II and 34.4% in Group III, consistent with the results of a similar study at Hebrew University, in which a 33% reduction in BMI was observed in one year following LGS [29].

The prevalence of OSAS in our study population was likely higher than our reported prevalence, which explains why we considered the patients positive who were screened at the appropriate center and diagnosed with OSAS—and, if necessary, undergoing therapy.

Postoperative mortality did not occur in our patient population, and significant morbidity was observed in four cases during the postoperative period. One of our patients underwent reoperation on account of suture failure due to peritonitis, a septic condition, followed by admission to the intensive care unit. Suture failure was presumably due to the patient’s serious diet error during the early postoperative period. Two cases required reoperation due to bleeding complications, and one case of postoperative pneumonia, in which the patient responded well to antibiotic treatment; the latter three patients did not require ITO admission. It corresponds with a severe mortality rate of 2.4%, a result which correlates with international data (based on the data of 3983 LSG analyzed in the study by Singhal et al., the mortality rate was 0.1%, and the incidence of severe complications (Clavien-Dindo 3–4) was 2.1% [30]).

Hence, our results are similar to international data, with moderate differences. However, it should be noted that the methodology of the cited studies is not entirely the same as ours (patient composition, meta-analysis versus single-center follow-up, etc.), so an exact comparison cannot be made.

The analysis of our patient population suggests, in consideration of severe obesity, advancing the age of our patients does not lead to a further increase in body weight, and they typically reach their ‘maximum’ body weight at a younger age (examining the average age of our Group I, II and III patients, the lowest average age can be observed even in Group III).

In the case of our patients who underwent LSG surgery, normal laboratory parameters, normal respiratory function and cardiac ultrasound findings, the absence of multimorbidity and the very low postoperative morbidity rates without mortality is primarily the consequence of the younger age among our patients (mean age, 44 and a half years).

Our results pose additional questions, which point to possible ways forward regarding future research. Our study did not investigate the proportion of patients with androgenic and gynoid obesity in our study population, the distribution of comorbidities in the two types of obesity patterns, nor whether there is a difference in the outcome of surgery between patients with androgenic and gynoid obesity, either in terms of weight loss or improvement in comorbidities. In the same context, examining the question in the male-female relation will prove helpful.

It will also be worthwhile to carry out targeted studies in the direction of OSAS since, for the reasons detailed earlier, our study is undoubtedly of limited value in this respect.

The present study was carried out in the form of a one-year follow-up; thus, our results are short-term results, while long-term effectiveness will be judged following 5 or 10 years of follow-up since patients need to maintain their lifestyle changes and diet to effectively sustain the results. To assess long-term morbidity and mortality, long-term follow-up of patients will also prove worthwhile.

## 5. Conclusions

Based on the analysis of the preoperative and postoperative results of our patients undergoing LSG, we conclude that a small proportion of patients had known comorbidities related to obesity, except hypertension and diabetes, the only significant comorbidities in our patient population (at the same time, we recommend more extensive testing for OSAS). 

We also conclude that the increase in body weight is not related to the advance in age among our patients. 

It is also important to note that surgery was effective in weight loss for all three groups in a short term (one year follow-up). 

A further important finding is that our morbidity and mortality figures are very favorable due to the younger age among patients, the low rate of comorbidities and the absence of multimorbidity.

Our results suggest LGS surgery is an effective weight loss method in younger patients without gross abnormalities in laboratory parameters, respiratory function and cardiac ultrasound findings, and without multimorbidity. LGS surgery also reduces the severity of hypertension in a significant proportion of patients with hypertension as a comorbidity and can be performed with very low perioperative morbidity and mortality rates.

LGS surgery proves beneficial in many ways. From the patient’s point of view, if the patient undergoes surgery at a younger age, more years of good quality of life can be enjoyed [31], and these years will be added to the patient’s younger, more active and productive life. From this respect, demographic factors are not negligible either, since our younger, obese patients about to bear children may face fewer problems during conception following a significant weight loss, and females may also face reduced maternal and fetal risk during pregnancy and labor [31].

Over the short term, from a health care system perspective, apart from obesity, younger patients with few comorbidities, no multimorbidity and good general health are on the operating table in hopes fewer complications can be expected from a surgical and anesthetic point of view. Shorter hospital stays, fewer post-operative complications, such as thromboembolic complications, pneumonia, decubitus, suture failure, wound infection, (early) reoperation and many other positive factors can be anticipated, pointing toward surgery at a younger age.

From the point of view of the healthcare system, over the long term, preventing the development of comorbidities, particularly multimorbidity, will avoid further strain upon the health system.

Regarding healthcare financing, the positive financing effects of lower morbidity and mortality rates, operations performed with shorter hospital stays, and the financial resources not spent on healthcare due to untreated multimorbidity can be reallocated to other realms of the healthcare system.

From an economic point of view, the complications mentioned earlier of obesity are associated with a reduction in vitality and, in many cases, with a partial or considerable reduction in working capacity and even total incapacity to work due to the significant lifestyle constraints and health problems of the individuals concerned. In addition to the negative impact on the individual’s well-being, the condition also impacts society as a whole. From an economic perspective, it is also desirable to avoid a significant reduction in the earning capacity of patients due to obesity and comorbidities or the patient becoming completely incapacitated. The trivial aspect of this is that the patient will then be dependent on public care for treatment, which, in the case of years of therapy, increases the cost to the public purse, whereas, in the absence of a health problem, the budget can be used more efficiently. On the one hand, the financial burden of managing comorbidities and multimorbidity in public and private health care is not negligible, as is the additional strain on the always scarce capacity of health care workers. Furthermore, the loss of gainful employment for a citizen of working age who is unable to work or has a reduced capacity to work due to a health condition must also be taken into account, both on the production side and in terms of the associated loss of tax revenue. As a consequence, following successful weight loss, if the ability to work is restored (or does not subsequently decline) as the health condition improves, not only is the budgetary expenditure on care for the individual concerned reduced but the overall economic contribution is also increased by the patient’s ability to work or working more efficiently.

Based on these results, it may be advisable to direct some patients who do not exhaust the classic indication for bariatric surgery towards the surgical solution: our results suggest it is not worth waiting for comorbidities, especially multimorbidity, to appear. 

## Figures and Tables

**Figure 1 medicina-59-02092-f001:**
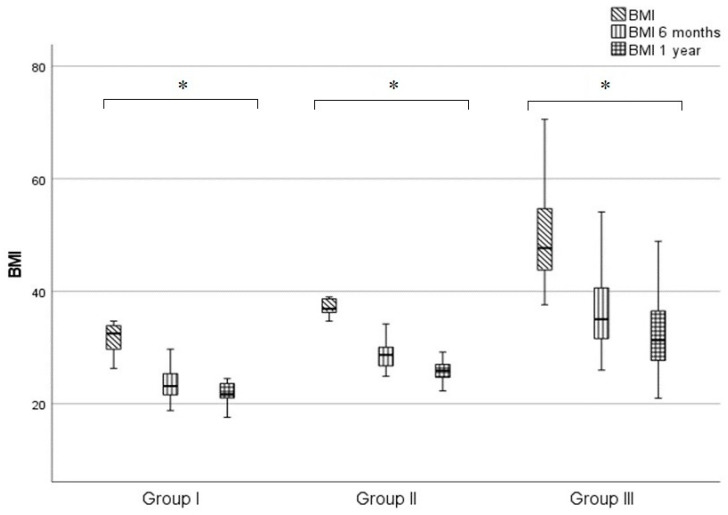
BMI values of each group prior to surgery, 6 months and one year following surgery. Data are plotted as “box plot”, median, interquartile range, minimum-maximum. * *p* < 0.001. Abbreviations: BMI: BMI prior to surgery, BMI 6 months: BMI six months following surgery, BMI 1 year: BMI one year following surgery.

**Figure 2 medicina-59-02092-f002:**
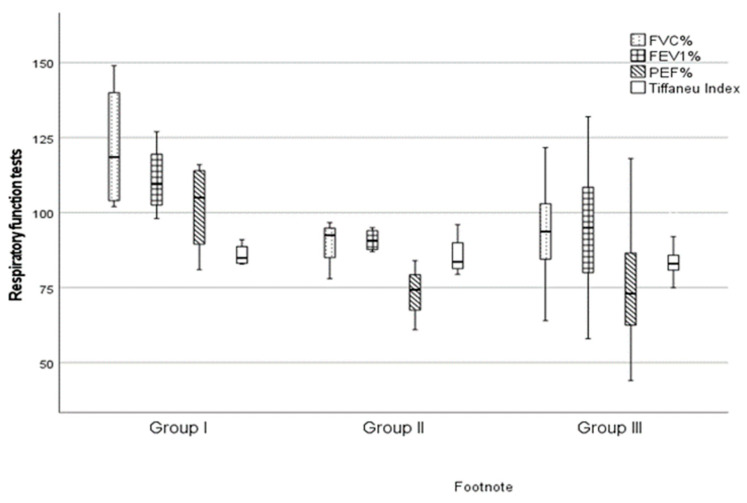
Results of patients’ preoperative respiratory function tests. Data are plotted as “box plot”, median, interquartile range, minimum-maximum. Abbreviations: FVC = Forced Vital Capacity, FEV1 = Forced Expiratory Volume, PEF = Peak Expiratory Flow.

**Table 1 medicina-59-02092-t001:** Patient demographics.

	Group I	Group II	Group III
Male/Female	4/18	8/20	54/60
Age (year)	45 (25–53)	43 (20–66)	38 (18–70)
Height (cm)	172 (161–182)	168 (141–190)	173 (150–200)
Weight (kg)	94 (70–138)	107 (83–146)	143(100–207)
BMI (kg/m^2^)	32 (30–34.7)	37 (35.6–39.9)	47 (40.1–76)

The data are described as median, minimum, and maximum.

**Table 2 medicina-59-02092-t002:** Ultrasound findings of patients.

	Group I	Group II	Group III	*p*
Left ventricle diameter	49 ± 3.91	49.5 ± 4.59	50 ± 4.57	0.165
Right ventricle diameter	27 ± 3.62	26 ± 3.33	30 ± 4.39	0.009
Ejection Fraction %	64 ± 5.61	62.5 ± 7.35	59 ± 5.55	0.072
E/A	1.21 ± 0.41	1.14 ± 0.36	1.17 ± 0.84	0.535

Data are presented as median and standard deviation. Abbreviations: E/A: early to atrial filling velocity ratio.

**Table 3 medicina-59-02092-t003:** Preoperative laboratory parameters.

	Group I	Group II	Group III	*p*
Cholesterol (mmol/L)	5.8 ± 0.66	5.51 ± 1.44	5.60 ± 1.14	0.722
LDL cholesterol (mmol/L)	3.49 ± 0.83	2.27 ± 1.15	3.36 ± 1.19	0.234
HDL cholesterol (mmol/L)	1.24 ± 0.07	1.38 ± 0.31	1.38 ± 1.09	0.797
Blood sugar (mmol/L)	4.09 ± 0.29	5.05 ± 1.68	5.05 ± 1.48	0.08
Hemoglobin A1c	5.04 ± 0.18	5.5 ± 0.55	5.90 ± 0.41	0.119
Cortizol (nmol/L)	358.70 ± 175.31	229.30 ± 164.07	227.20 ± 93.31	0.866
TSH (mIU/L)	1.92 ± 1.30	1.51 ± 0.95	2.19 ± 1.13	0.366

Data are presented as median and standard deviation. Abbreviations: LDL = Low Density Lipoprotein, HDL = High Density Lipoprotein TSH = Thyroid Stimulating Hormone.

**Table 4 medicina-59-02092-t004:** Comorbidities of patients.

	Group I	Group II	Group III	*p*
HT prior to surgery (%)	51	43	45	0.368
HT improved following surgery in HT patients (%)	36	34	40	0.669
DM II prior to surgery (%)	11	11	15	0.611

Abbreviations: HT: hypertension, DM: Type II Diabetes Mellitus.

## Data Availability

The patient’s data is available in appropriate official hospital electronic data storage systems.

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
