# Peer review of "Indications for Sleeve Gastrectomy—Is It Worth Waiting for Comorbidities to Develop?"

_medicina, 2023, doi:10.3390/medicina59122092_

Round 1
Reviewer 1 Report
Comments and Suggestions for Authors
In this manuscript, the authors evaluated the effect of sleeve gastrectomy on obesity, as well as the comorbidities, in patients undergoing LSG. This manuscript is well-written and the information could be helpful to researchers and doctors in decision making. However, some concerns are raised about the data presentation and interpretation, which might affect the quality of this manuscript and lead to confusion.
1) In Figure 1, please add significance levels by asterisks to indicate the p values of the U-test. Meanwhile, it would be helpful to add an axis title to the left y-axis.
2) In Table 1, the maximum BMI of Group 1 and Group 2 are 43 and 52, respectively. However, as indicated in Materials and Methods (Line 113-114), members of Group 1 had a BMI between 30-34.9 and Group 2 between 35-39.9kg/m2. It would be great if the authors could explain why the maximum values in Table 1 exceed the maximum values indicated in Materials and Methods.
3) In the Results Section, the authors state in Lines 136-137 that "no statistically verifiable difference was found". Please specify the method used to verify the difference. If the authors mean the difference is not significant, it would be more appropriate to say that the difference does not reach statistical significance.
4) Please add the axis title to the left y-axis of Figure 2.
5) In Line 148, the authors state "hypertension patients improved". Please give a detailed explanation and data to support and clarify this statement.
6) In Lines 154-155, the authors conclude that the increase in body weight is not related to the aging of our patients. It would be great if the authors cold provide evidence that supports this conclusion. Has any correlation analysis or other type of statistical analysis been done to address this hypothesis? If the answer is yes, please include it in the manuscript.
Author Response
Dear Reviewer!Thank you very much for reviewing my manuscript. Please see my answers to your questions. I hope you will be satisfied.
1) In Figure 1, please add significance levels by asterisks to indicate the p values of the U-test. Meanwhile, it would be helpful to add an axis title to the left y-axis.
Answer: I made the requested change.
2) In Table 1, the maximum BMI of Group 1 and Group 2 are 43 and 52, respectively. However, as indicated in Materials and Methods (Line 113-114), members of Group 1 had a BMI between 30-34.9 and Group 2 between 35-39.9kg/m2. It would be great if the authors could explain why the maximum values in Table 1 exceed the maximum values indicated in Materials and Methods.
Answer:
ANSWER: Dear reviewer. Your suggestion is fully justified,presumably the source of the error was a typo,
the authors apologize for their obvious mistake.
In connection with this, however, we checked our calculations and,
contrary to our previous finding, we notice a significance in one respect:
in group III the right ventricle diameter is significantly higher
than in the other groups, however,
this difference remains within the physiological limits
and does not affect our conclusions. We apologize once again for the mistake, corrections are made
3) In the Results Section, the authors state in Lines 136-137 that "no statistically verifiable difference was found". Please specify the method used to verify the difference. If the authors mean the difference is not significant, it would be more appropriate to say that the difference does not reach statistical significance.
Answer: I made the requested change.
4) Please add the axis title to the left y-axis of Figure 2.
Answer: I made the requested change.
5) In Line 148, the authors state "hypertension patients improved". Please give a detailed explanation and data to support and clarify this statement.
Answer: I made the requested change. I hope you find the change appropriate.
6) In Lines 154-155, the authors conclude that the increase in body weight is not related to the aging of our patients. It would be great if the authors cold provide evidence that supports this conclusion. Has any correlation analysis or other type of statistical analysis been done to address this hypothesis? If the answer is yes, please include it in the manuscript.
Answer: Sperman's correlation analysis was performed, and it confirmed that body weight is not related to the aging of our patients. (p=0.082). It is also clearly visible without statistics that the average age of III. the lowest in the group.
Reviewer 2 Report
Comments and Suggestions for Authors
Dear authors,
Thank you very much for submitting your manuscript to be prestigious journal Medicina.
I hope that my suggestions and remarks will be useful in order to increase the quality of your paper.
1. Lines 8-18 – please use only professional e-mails instead of personal ones.
2. Lines 49-66 – There are no citations to support your statements.
3. Line 175 – In tables 2-4 is it necessary to mention p value if in every row you mentioned NS?
4. Line 189 – The Discussion section is, from my point of view, insufficient and requieres significant elaboration as there exists a lot of literature on this topic.
5. Line 230 – The entire Conclusions section should be rephrased in a much synthetic manner focused on the clinical/practical impact of your finding on controlling obesity and its’s side effects.
Please receive my best regards!
Comments on the Quality of English LanguageMDPI English and layout editing services are highly recommended.
Author Response
Dear Reviewer
Thank you very much for reviewing my manuscript. Please see my answers to your questions.
I hope you will be satisfied.
1, Lines 8-18 – please use only professional e-mails instead of personal ones.
Answer: I made the requested change.
- Lines 49-66 – There are no citations to support your statements.
Answer: I made the requested change, added citations to support my statements.
- Line 175 – In tables 2-4 is it necessary to mention p value if in every row you mentioned NS?
Answer: I made the requested change, I mentioned p values.
- Line 189 – The Discussion section is, from my point of view, insufficient and requieres significant elaboration as there exists a lot of literature on this topic.
Answer: I made the requested change. I did a significant elaboration, based on exist lterature.
- Line 230 – The entire Conclusions section should be rephrased in a much synthetic manner focused on the clinical/practical impact of your finding on controlling obesity and its’s side effects.
Answer: I made the requested change, I rephased and extended the whole Conclusion, I focused on he clinical/practical impact of our findings.
I really hope you will find my answers satisfactory. Sincerely,
Round 2
Reviewer 1 Report
Comments and Suggestions for Authors
Thank you very much for the revision and response. This revised version of the manuscript has a greatly enhanced quality of the Introduction, Results, and Discussion chapter.
Here are some minor issues that need to be taken care of.
1. There is a typo in Table 1 as the authors indicate the maximum BMI for Group 3 is "1-76".
2. In Figure 1, the authors used different numbers of asterisks to indicate the same p-value level, which is inappropriate.
Author Response
Dear Reviewer!
Thank you again for taking the time to review my manuscript.
Allow me to answer your questions!
- There is a typo in Table 1 as the authors indicate the maximum BMI for Group 3 is "1-76".
Answer: I checked again the version I submited, and the version the Journal sent me back allso, in all these 2 documents I see „47 (40,1-76)”, as 47 as median, 40,1 as minimum, and 76 as maximum of BMI in case of Group III patients in the last column in Table 1, so I don’t understand why the reviwer saw „1-76”. I hope you will see the correct numbers in the version I submit this time. I will highlite it with yellow.
- In Figure 1, the authors used different numbers of asterisks to indicate the same p-value level, which is inappropriate.
Answer: In accordance with your request, I modified Figure 1 and inserted it into the manuscript, I changed the „* p<0.001, ** p<0.001, *** p<0.001,” to „* p<0.001” under Figure 1.
I hope you will find everything in order this time.
Respectfuly, Miklós Siptár
Reviewer 2 Report
Comments and Suggestions for Authors
The authors have addressed properly the remarks.
Comments on the Quality of English LanguageMinor English editing is required.
Author Response
Dear Reviewer!
You mentioned that Minor English editing is required.
Answer: A native language instructor of University of Pécs, Meical Scool, Department of Languages for Biomedical Purposes and Communication, linguistically proofread and corrected the manuscript, and gave us a Certificate, which I am attaching. Linguistical corrections are marked by comments and makeups in the manuscript.
I hope you will find everything in order this time.
Respectfully,
Miklós Siptár
